# Rosmarinic Acid Increases Macrophage Cholesterol Efflux through Regulation of ABCA1 and ABCG1 in Different Mechanisms

**DOI:** 10.3390/ijms22168791

**Published:** 2021-08-16

**Authors:** Jean-Baptiste Nyandwi, Young Shin Ko, Hana Jin, Seung Pil Yun, Sang Won Park, Hye Jung Kim

**Affiliations:** 1Department of Pharmacology, Institute of Health Sciences, College of Medicine, Gyeongsang National University, Jinju 52727, Korea; nbaptiste1988@gmail.com (J.-B.N.); shini33@naver.com (Y.S.K.); hanajin.kr@daum.net (H.J.); spyun@gnu.ac.kr (S.P.Y.); parksw@gnu.ac.kr (S.W.P.); 2Department of Convergence Medical Science (BK21 Plus), Gyeongsang National University, Jinju 52727, Korea; 3Department of Pharmacy, School of Medicine and Pharmacy, College of Medicine and Health Sciences, University of Rwanda, Kigali 4285, Rwanda

**Keywords:** ABCA1, ABCG1, cholesterol efflux, diabetic atherosclerosis, macrophages, rosmarinic acid

## Abstract

Lipid dysregulation in diabetes mellitus escalates endothelial dysfunction, the initial event in the development and progression of diabetic atherosclerosis. In addition, lipid-laden macrophage accumulation in the arterial wall plays a significant role in the pathology of diabetes-associated atherosclerosis. Therefore, inhibition of endothelial dysfunction and enhancement of macrophage cholesterol efflux is the important antiatherogenic mechanism. Rosmarinic acid (RA) possesses beneficial properties, including its anti-inflammatory, antioxidant, antidiabetic and cardioprotective effects. We previously reported that RA effectively inhibits diabetic endothelial dysfunction by inhibiting inflammasome activation in endothelial cells. However, its effect on cholesterol efflux remains unknown. Therefore, in this study, we aimed to assess the effect of RA on cholesterol efflux and its underlying mechanisms in macrophages. RA effectively reduced oxLDL-induced cholesterol contents under high glucose (HG) conditions in macrophages. RA enhanced ATP-binding cassette transporter A1 (ABCA1) and G1 (ABCG1) expression, promoting macrophage cholesterol efflux. Mechanistically, RA differentially regulated ABCA1 expression through JAK2/STAT3, JNK and PKC-p38 and ABCG1 expression through JAK2/STAT3, JNK and PKC-ERK1/2/p38 in macrophages. Moreover, RA primarily stabilized ABCA1 rather than ABCG1 protein levels by impairing protein degradation. These findings suggest RA as a candidate therapeutic to prevent atherosclerotic cardiovascular disease complications related to diabetes by regulating cholesterol efflux in macrophages.

## 1. Introduction

Atherosclerosis is the key driver of vascular diseases such as stroke, angina, myocardial infarction and brain ischemia in diabetes and obesity [1,2]. Lipid dysregulation in diabetes mellitus escalates endothelial inflammation and dysfunction, critical events involved in atherogenesis [3,4]. Monocytes are primordial inflammatory cells in early atherosclerotic plaques that have been reported to play an integral role in atherosclerosis progression [5]. Monocytes interact with endothelial cells (ECs), after which monocytes migrate in the subendothelial area of the vessel wall and differentiate into macrophages, where they take up cholesterol from retained lipoproteins and transform into foam cells, which play a critical role in the progression of atherosclerosis [6,7]. Macrophage lipid homeostasis depends on cholesterol engulfment and free cholesterol efflux, which are mediated by multiple molecules [8]. Overexpression of the scavenger receptors (SRs) such as cluster of differentiation (CD) 36 or lectin-like receptor for oxidized low-density lipoproteins (LOX-1) on macrophages triggers dysregulated cholesterol uptake, leading to foam cell formation in the subendothelial area [9]. Conversely, macrophage foam cell formation is inhibited by cholesterol efflux from macrophages and following metabolism by the liver [10,11].

Macrophage cholesterol efflux and reverse cholesterol transport (RCT) substantially protect against the progression of atherosclerosis [12,13]. ATP-binding cassette transporter A1 (ABCA1) and G1 (ABCG1) mediate cholesterol efflux from foam cells to the extracellular cholesterol acceptors apolipoprotein A-I (ApoA-I) and high-density lipoprotein (HDL), respectively, thereby limiting foam cell formation and decreasing the progression of atherosclerosis [14,15]. Previous studies demonstrated that atherosclerosis-prone LDL receptor-deficient mice that were transplanted with bone marrow from ABCA1- and ABCG1-knockout mice showed accelerated atherosclerotic lesion development [16], whereas mice that were transplanted with bone marrow from ABCA1-overexpressing mice exhibited decreased atherosclerosis progression [17]. Consistently, ablation of macrophage ABCA1 and ABCG1 in mice resulted in a substantial increase in atherosclerosis progression [18]. In addition, transgenic mice overexpressing ABCA1 showed a low incidence of atherosclerosis [19]. Moreover, studies in humans have reported that cholesterol efflux capacity is negatively correlated with atherosclerotic cardiovascular events [20,21,22]. Such evidence sheds light on the imperative roles of endothelial dysfunction, foam cell formation and cholesterol efflux in atherogenesis. Regarding the ABCA1 and ABCG1 induction mechanisms, liver X receptor (LXR) agonists are the most well-identified transcription factors to stimulate ABCG1 and ABCA1 expression, resulting in promotion of cholesterol efflux from macrophages, and eventually protection against atherosclerosis in mice [15,23]. Moreover, it has been known that Janus kinase (JAK) activated signal transducer and activator of transcription 3 (STAT3) signaling pathway increased ABCA1 and ABCG1 levels in macrophages [24,25].

Rosmarinic acid (RA) is a natural phenolic ingredient found in *Rosmarinus officinalis* and other common culinary herbs from the Lamiaceae family [26]. Many studies have asserted that RA possesses antioxidant, anti-inflammatory, antidiabetic and cardioprotective activities [27,28]. We previously demonstrated that RA inhibits diabetic endothelial dysfunction by inhibiting inflammasome activation in ECs through downregulating the p38-FOXO1-TXNIP pathway [29]. However, the effect of RA on cholesterol efflux by macrophage remains unknown. Therefore, in this study, we aimed to investigate the atheroprotective effect of RA through enhancing cholesterol efflux and reducing cholesterol levels in macrophage. In particular, we have revealed the molecular mechanism associated with RA-mediated ABCA1 and ABCG1 induction.

## 2. Results

### 2.1. RA Reduced oxLDL-Induced Lipid and Cholesterol Contents under High Glucose (HG) Conditions in Macrophage

Under HG conditions, macrophages in the subendothelial area engulf oxLDL and transform into lipid-leaden macrophages or foam cells, which accelerate the inflammatory process and plaque formation during atherogenesis [30]. Hence, we examined whether RA could reduce lipid contents and cholesterol levels in macrophages treated with oxLDL under high glucose conditions. When we evaluated the effect of RA on macrophage viability, RA was not toxic to macrophages at 100 μM for up to 72 h (Figure 1A,B). When THP-1 macrophages were stimulated with oxLDL under low glucose conditions, oxLDL treatment increased lipid accumulation, which was more enhanced in high glucose conditions (Figure 1C). HG- and oxLDL-induced lipid contents in THP-1 macrophages were reduced in the presence of RA (Figure 1C). We further examined the effect of RA on the cholesterol content of THP-1 macrophages. Fractionation analysis of the intracellular cholesterol profile demonstrated that RA dose-dependently decreased the intracellular total cholesterol content (Figure 1D), free cholesterol content (Figure 1E) and cholesteryl ester content (Figure 1F) in THP-1 macrophages treated with oxLDL under HG conditions. Because lipid uptake by macrophages is induced by SRs expressed on macrophages, we evaluated the effect of RA on the key SRs responsible for oxLDL uptake by macrophages, LOX-1 and CD36. RA treatment decreased LOX-1 and CD36 (Figure 1G,H) expression, which was increased by oxLDL under HG conditions, in a dose-dependent manner. These data demonstrate that RA effectively reduced lipid contents and macrophage foam cell formation by decreasing lipid uptake, which were enhanced by oxLDL under HG conditions.

### 2.2. RA Increases Cholesterol Efflux by Upregulating ABCA1 and ABCG1

Research findings suggested that ABCA1 and ABCG1 are key transporters that facilitate cholesterol egress from macrophages, reducing atherosclerosis development [14]. We then explored the effect of RA on the protein expression levels of ABCA1 and ABCG1 in THP-1 macrophages stimulated with oxLDL under HG conditions. oxLDL treatment did not affect ABCA1 or ABCG1 protein expression levels in THP-1 macrophages maintained under HG conditions. Interestingly, RA treatment dose-dependently increased ABCA1 (Figure 2A) and ABCG1 (Figure 2B) protein levels in THP-1 macrophages. Then, we assessed whether the RA-mediated increases in the ABCA1 and ABCG1 proteins impacted cholesterol efflux efficiency in THP-1 macrophages. RA significantly increased cholesterol efflux to the extracellular cholesterol acceptors ApoA1 (Figure 2C) and HDL (Figure 2D) in THP-1 macrophages in a dose-dependent manner, suggesting that RA increases cholesterol efflux, resulting in a decrease in foam cell formation.

### 2.3. RA Mediated ABCA1 and ABCG1 Expression through the Activation STAT3 Signaling in THP-1 Macrophages

Next, we explored the mechanisms by which RA induces ABCA1 and ABCG1 expression. Previous studies have shown that JAK2/STAT3 signaling is involved in the modulation of ABCA1 and ABCG1 expression [25,31]. Thus, we investigated whether the STAT3 pathway is also involved in RA-mediated ABCA1 and ABCG1 expression. Pretreatment with AG-490 (a specific JAK2/STAT3 inhibitor) significantly abrogated the RA-mediated increase in ABCA1 (Figure 3A) and ABCG1 (Figure 3B) expression. Then, we confirmed that treatment with RA activated STAT3 in a time-dependent manner, with the maximum level observed 4 h after RA treatment (Figure 3C), which was also inhibited in the presence of AG-490 (Figure 3D). These results suggest that RA induced ABCA1 and ABCG1 protein expression in THP-1 macrophages by activating the JAK2/STAT3 pathway.

### 2.4. RA Induced ABCA1 and ABCG1 Expression through Differential Regulation of the MAPK Pathways in THP-1 Macrophages

In addition to our assessment of the JAK2/STAT3 pathway, we determined the involvement of MAPKs in the RA-mediated increase in ABCA1 and ABCG1 expression in THP-1 macrophages. We pretreated cells with a p38 inhibitor (SB203580), a JNK inhibitor (SP600125) and an ERK inhibitor (PD98059) for 1 h and then treated them with RA for 24 h. The RA-mediated increase in ABCA1 protein expression was decreased by the p38 and JNK inhibitors but not by the ERK1/2 inhibitor. However, the RA-induced increase in ABCG1 protein levels was decreased by all inhibitors: the p38, JNK and ERK inhibitors (Figure 4A). Furthermore, we assessed whether RA activates the MAPK and PKC signaling pathways. We treated THP-1 macrophages with RA and harvested them after 0.5, 1, 2, 4 and 8 h to determine the levels of phospho-ERK1/2, phospho-p38 and phospho-JNK. As shown in Figure 4B–D, RA significantly increased the phosphorylation of ERK1/2 (Figure 4B), p38 (Figure 4C) and JNK (Figure 4D) beginning at 30 min until 8 h.

### 2.5. RA-Activated PKC Pathway Was Involved in the ABCA1 and ABCG1 Expression through Differential Regulation of p38 and ERK1/2 Pathway

Moreover, we investigated the role of PKC on the RA-mediated induction of ABCA1 and ABCG1 expression. Results showed that induction of ABCA1 and ABCG1 expression by RA was significantly suppressed by GF109203X (5 μM), a PKC inhibitor (Figure 5A), and RA effectively activated PKC pathway from 15 min (Figure 5B). RA-mediated PKC phosphorylation at 15 min was inhibited in the presence of GF109203X (Figure 5C). Interestingly, the activation of PKC seemed to precede the RA-mediated activation of MAPK, so we tested whether activation of the PKC pathway by RA would affect the MAPK pathway in THP-1 macrophages. We found that RA-activated ERK1/2 (Figure 5C) and p38 phosphorylation (Figure 5D) was significantly reduced by pretreatment with GF109203X, a specific PKC inhibitor. However, RA-induced JNK activation was not affected by GF109203X (Figure 5E). These findings suggest that RA differentially regulates ABCA1 and ABCG1 expression levels; in THP-1 macrophages, RA regulated ABCA1 expression through PKC-p38 MAPK and ABCG1 expression through PKC-ERK1/2/p38 MAPK. Furthermore, RA-activated JNK pathway was involved in the ABCA1 and ABCG1 regardless of PKC pathway.

### 2.6. RA Stabilized ABCA1 Primarily Rather Than ABCG1 Protein Levels by Impairing Protein Degradation in THP-1 Macrophages

Lastly, we further examined whether RA increases ABCA1 and ABCG1 protein levels by affecting protein degradation. We treated THP-1 macrophages with RA for 24 h to allow ABCA1 and ABCG1 protein synthesis and then blocked de novo protein synthesis by applying CHX (140 μM) for different durations (10, 20, 40 and 180 min). The ABCA1 protein level was significantly reduced from 10 min and showed less than 50% the level in the control group at 3 h after CHX application (Figure 6A). However, the ABCG1 protein level was not changed at early times and showed the significant decrease at the later time of 180 min after CHX application (Figure 6B). In the presence of RA, the ABCA1 and ABCG1 protein levels was not decreased even in the presence of CHX (Figure 6A,B). In other words, RA treatment significantly restored the ABCA1 and ABCG1 protein levels, mainly ABCA1, even in the presence of CHX. According to reports, ABCA1 and ABCG1 protein degradation involves the proteasome-, lysosome- and calpain-mediated protein degradation pathways [32]. Therefore, we determined the protein degradation pathway through which RA regulates ABCA1 and ABCG1 protein levels using specific inhibitors of these protein degradation pathways: the proteasome inhibitor lactacystin, the lysosome inhibitor chloroquine and the calpain inhibitor calpeptin. Lactacystin, a proteasome inhibitor, and chloroquine, a lysosome inhibitor, augmented ABCA1 and ABCG1 protein levels, as did RA (Figure 6C,D). However, the protein levels of ABCA1 and ABCG1 remained unchanged upon cotreatment with either lactacystin or chloroquine and RA (Figure 6C,D), suggesting that the ability of RA to induce ABCA1 and ABCG1 protein expression is due to inhibition of the lysosome- or proteasome-mediated degradation pathways. In Figure 6E, the calpain inhibitor calpeptin increased ABCA1 and ABCG1 protein expression, similar to the effects of RA, and cotreatment with calpeptin and RA did not affect ABCG1 protein level, but interestingly, significantly enhanced ABCA1 protein level. These results suggest that RA stabilized ABCA1 and ABCG1 protein levels through different mechanisms; ABCA1 by impairing proteasome- and lysosome-mediated degradation and ABCG1 by impairing proteasome-, lysosome- and calpain-mediated degradation in THP-1 macrophages.

## 3. Discussion

Several studies have reported that hyperglycemia is associated with dyslipidemia and increases the risk of atherosclerotic vascular diseases in diabetes [1,2,33]. Endothelial dysfunction is regarded as the initial step related to atherogenesis under hyperglycemic and dyslipidemic conditions [34]. We previously showed that oxLDL induced inflammasome activation in ECs under HG conditions, but RA effectively inhibited inflammasome activation and the resulting endothelial dysfunction by downregulating the p38-FOXO1-TXNIP pathway [29].

The increased uptake of lipoproteins and reduced cholesterol efflux promote the formation of cholesterol-laden macrophage foam cells, which accelerate the development of atherosclerotic lesions and plaques [35]. Macrophages limit foam cell formation by converting cholesteryl ester into free cholesterol through hydrolyzing modified LDL that has been taken up, or through lipolysis and lipophagy of lipid droplets inside the cells. Moreover, cells modulate the cytotoxic effect of free cholesterol by subjecting free cholesterol to RCT or esterifying surplus free cholesterol back to cholesteryl ester [36,37]. Therefore, in this study, we examined the effect of RA on macrophage-mediated lipid homeostasis, lipid uptake and cholesterol efflux. Figure 1C–F shows that RA significantly decreased the oxLDL-mediated increase in intracellular lipid content, total cholesterol and cholesteryl esters under HG conditions in a dose-dependent manner. Interestingly, the cholesteryl ester and total cholesterol contents were significantly reduced by RA at a low dose of 1 µM (Figure 1D,F), but a significant decrease in free cholesterol content was observed upon treatment with RA at concentrations beginning at 50 µM (Figure 1E). Because the expression of SRs such as CD36 and LOX-1, which mediate the recognition and engulfment of oxLDL, is increased under hyperglycemic and dyslipidemic conditions and mediates the effect of oxLDL [30,38], we further examined whether RA could reduce SR expression and subsequently impair oxLDL-mediated lipid uptake by macrophages under HG conditions. Figure 1G and 1H showed that RA significantly decreased expression of the SRs LOX-1 and CD36 induced by oxLDL under HG conditions, suggesting that RA reduces the oxLDL-mediated increase in intracellular lipid content, total cholesterol and cholesteryl esters under HG conditions by reducing expression of the SRs LOX-1 and CD36 induced by oxLDL under HG conditions.

An enormous body of evidence has indicated that cholesterol efflux from macrophages is the main process that inhibits atherosclerosis development [22,39]. Cholesterol efflux is achieved through extracellular cholesterol acceptors such as ApoA1 and HDL, which require the activity of membrane transporters such as ABCA1 and ABCG1 [40,41]. We therefore investigated whether RA could promote cholesterol efflux from macrophages by increasing ABCA1 and ABCG1 expression. We found a significant dose-dependent increase in ABCA1 and ABCG1 expression (Figure 2A,B) and ApoA1- and HDL-mediated cholesterol efflux upon RA treatment in macrophages (Figure 2C,D). According to reports, ABCG1 directly supplies free cholesterol to mature HDL, whereas ABCA1 preferentially conveys free cholesterol to lipid-poor ApoA1 to form nascent, immature HDL molecules, which are in turn matured by the lecithin-cholesterol acyl transferase enzyme, hence increasing the number of functional HDL molecules [42,43]. Our findings suggest that RA is an efficient modulator of cholesterol efflux to ApoA1- and HDL acceptor.

Next, we explored the mechanism involved in RA-driven ABCA1 and ABCG1 protein induction. Generally, protein levels are the result of a balance between de novo synthesis and degradation. First, we investigated the signaling pathways pertain to RA-induced ABCA1 and ABCG1 upregulation. Our results showed that RA effectively activated the STAT3 pathway, and RA-induced ABCA1 and ABCG1 expression was significantly inhibited by the selective JAK2/STAT3 inhibitor AG490 (Figure 3A,B), suggesting that RA induced ABCA1 and ABCG1 expression through activation of the JAK/STAT3 signaling pathway in THP-1 macrophages, which is consistent with other reports [24,31]. In addition, we investigated the role of MAPKs and PKC pathways in the RA-induced increase in ABCA1 and ABCG1. p38 MAPK upregulates ABCA1 expression in macrophages through sp1- and LXR-binding sites in the ABCA1 promoter region [44], and ERK1/2 pathway activation also increases ABCA1 expression [12]. Moreover, it has been reported that HSP70 downregulates ABCA1 and ABCG1 expression through inhibiting JNK and its downstream Elk-1 in macrophages [45]. Similar to these reports, RA activated the ERK1/2, p38, JNK and PKC pathways (Figure 4B–D and Figure 5B). However, interestingly, we found that PKC is an upstream signal of ERK1/2 and p38 but not JNK (Figure 5E) and that ERK1/2 is involved in ABCG1 induction but not ABCG1 induction by RA (Figure 4A), suggesting that RA induced ABCA1 through the activation of PKC-p38 MAPK and induced ABCG1 through the activation of PKC-ERK1/2/p38 MAPKs in THP-1 macrophages, and furthermore the RA-activated JNK pathway was involved in the ABCA1 and ABCG1 regardless of PKC pathway. Several important studies demonstrated that MAPK-mediated activation of downstream transcription factors resulted in the induction of ABCA1 and ABCG1 in macrophages. Tanshinone IIA activated ERK/Nrf2/HO-1 signaling pathway to mediate ABCA1 and ABCG1 induction in macrophages [46]. Similarly, p38 MAPK-activated sp1 interacted with LXRα to activate ABCA1 transcription in macrophages [44]. Moreover, the activation of the JNK signaling pathway stimulated activator protein-1 (AP-1) DNA binding activity to upregulate ABCA1 expression in macrophages [47]. On the other side, some studies reported ABCG1 degradation in response to p38 and JNK2 activation in macrophages [48,49]. Nagelin et al. reported that 12/15-lipoxygnease targets ABCG1 for serine phosphorylation and destabilization through p38- and JNK2-dependent pathways in murine macrophage [48,49]. We carefully hypothesize that the difference between our results and the previous works may be partly due to the use of different species of cells. Indeed, murine and human cells possess different ABCG1 isoforms, which present distinct post-translational processing. It is reported that ABCG1 (+12) isoform is expressed in human cells but not in mouse cells [50]. Notably, ABCG1 (+12) phosphorylation stabilized ABCG1 protein levels in human cells and facilitated ABCG1-dependent cholesterol efflux to HDL [51,52]. One more possible reason is that different stimuli may cause different outcomes at different time points, even if they come into contact with the same signaling molecule. In our previous study, oxLDL under HG conditions activated p38 phosphorylation, which was involved in the inflammasome activation [29], but oxLDL stimulation under HG conditions didn’t induce ABCA1 and ABCG1 expression in this study. However, RA activated p38 phosphorylation and significantly induced ABCA1 and ABCG1 expression through p38 pathway in this study. Thus, further investigation to elucidate the role of p38 and JNK pathways in human ABCG1 protein degradation is needed. In fact, STAT3 is activated by phosphorylation at both Tyr705 and Ser727 residues. STAT3Tyr705 phosphorylation is mediated by a wide variety of growth factors including IL-6. In response to IL-6, STAT3 is transiently associated with gp130 and subsequently phosphorylated by JAKs on Tyr705 of STAT3 [53]. In addition, STAT3 is a substrate for several protein kinases [54] including ERK1/2 MAPK [55], and PKC directly or indirectly phosphorylates STAT3Ser727 via association with other protein kinase (Raf1, MAPK/ERK1/2 (MEK1/2), ERK1/2, p38MAPK). In our study, we determined the STAT3Tyr705 phosphorylation by RA, so our results in this study do not suggest the role of PKC or other associated protein kinases in the RA-mediated STAT3 phosphorylation.

In addition, we investigated whether RA hampers the rate of ABCA1 or ABCG1 degradation. Upon RA treatment, ABCA1 and ABCG1 protein levels were sustained even in the presence of CHX, rather increased compared to control levels. Interestingly, CHX-mediated protein degradation and the recovery of RA was more prominent in the ABCA1 protein (Figure 6). These results suggest that RA increased the ABCA1 and ABCG1 protein levels, mainly ABCA1, by interfering with protein degradation.

Taken together, these findings suggest for the first time that RA reduces atherosclerotic foam cells and enhances cholesterol efflux from macrophages under HG conditions by inducing the transporters ABCA1 and ABCG1, and RA differentially regulates ABCA and ABCG1 (Figure 7). RA regulates ABCA1 and ABCG1 expression levels through activation of the JAK2/STAT-3 pathway. RA-activated PKC and MAPKs are differentially involved in the effect of RA on ABCA1 and ABCG1; RA induces ABCA1 through PKC-p38 MAPK and induces ABCG1 through PKC-ERK1/2/p38 MAPK in macrophages. Atherogenesis is a complex process that progresses from the development of fatty streaks to the formation of thrombus within the intima. Lipids uptake by macrophages and foam cells formation and subsequent fatty streak formation occurs at early stage of atherogenesis [56]. Moreover, defective cholesterol efflux from macrophages leads to free cholesterol-induced cytotoxicity and may promote macrophage death in advanced lesions [57]. Given that RA modulates macrophage foam cells’ formation and increases cholesterol efflux, RA would be beneficial at the early stage of atherogenesis. Further clinical studies are needed.

## 4. Materials and Methods

### 4.1. Materials

Rosmarinic acid (RA; R4033) was purchased from Sigma-Aldrich (St. Louis, MO, USA). RPMI-1640 medium, DMEM (low glucose, #SH30021.01; high glucose, #SH30243.01), fetal bovine serum (FBS), a 100X penicillin-streptomycin solution and 0.05% trypsin-EDTA were purchased from HyClone Laboratories (Logan, UT, USA). Phorbol 12-myristate 13-acetate (PMA), human low-density lipoprotein (LDL; #437644), human high-density lipoprotein (HDL; #LP3) and human ApoA1 (#178452) were provided by Merck Millipore (Burling, MA, USA). Antibodies against LOX-1 (ab60178), CD36 (ab133625), ABCA1 (ab18180) and ABCG1 (ab52617) were purchased from Abcam (Cambridge, UK). Antibodies against phospho-JNK (#9251)/JNK (#9252), phospho-p38 (#9211), phospho-PKC (#9375S) and phospho-STAT3 (#9145)/STAT3 (#4904) were purchased from Cell Signaling Technology (Beverly, MA, USA). Antibodies against p38 (sc-535), PKC (sc-10800) and phospho-ERK (sc-7383)/ERK (sc-94) were obtained from Santa Cruz Biotechnology (Dallas, TX, USA). Enhanced chemiluminescence (ECL) western blotting detection reagent was obtained from Bio-Rad (Hercules, CA, USA). Lactacystin was procured from Cayman Chemical (Ann Arbor, MI, USA). BCECF was obtained from Boehringer (Mannheim, Germany). PD98059, SP600125, AG490, Oil Red O (ORO), chloroquine diphosphate, calpeptin and an antibody against β-actin (#a2066) were purchased from Sigma-Aldrich (St. Louis, MO, USA). SB203580 and GF109203X were obtained from Tocris (Bristol, UK). All other reagents including cycloheximide (CHX; #C4859) were obtained from Sigma-Aldrich.

### 4.2. Preparation of Oxidized Low-Density Lipoprotein (oxLDL) and Cell Treatment

oxLDL was prepared as described by Ko et al. [58] In brief, human LDL was dialyzed against phosphate-buffered saline (PBS) for 16 h at 4 °C to remove EDTA and then oxidized with 5 μM CuSO_4_ for 16 h at 37 °C. Then, the reaction was stopped by the addition of 1 mM EDTA and incubated for 24 h at 4 °C. To treat cells with oxLDL at the concentration of 100 μg/mL, we added 50 μL of oxLDL stock solution (2 mg/mL) per 1 mL of media.

### 4.3. Cell Culture and Treatment

Human THP-1 monocytes were originally obtained from American Type Culture Collection (Manassas, VA, USA). THP-1 cells were grown in RPMI-1640 medium supplemented with 10% FBS, 100 IU/mL penicillin and 10 μg/mL streptomycin. Cells were maintained in a humidified incubator at 37 °C with 95% air and 5% CO_2_. The THP-1 cells were differentiated into macrophages by stimulation with 150 nM PMA for 24 h. Then, the differentiated cells were changed with low glucose DMEM media containing 1 g/L of D-Glucose (5 mM Glu) and 20 mM mannitol as control or high glucose DMEM media containing 4.5 g/L of D-Glucose (25 mM Glu) as the HG condition. Cells were used at the fifth through ninth passage for all the experiments. All kinase inhibitors and RA were dissolved in dimethyl sulfoxide (DMSO) to prepare stock solutions. The stock solutions were diluted in culture medium to the working concentration as indicated. The final concentration of DMSO was ~0.1% (*v/v*).

### 4.4. Cell Viability Assay

THP-1 macrophage were seeded in black 96-well plates and incubated until they reached confluence. The confluent cells were treated with or without RA (100 µM) for the indicated time. After treatment, cell viability was measured by the cell counting kit-8 (CCK-8; Dongin biotech, Seoul, Korea) as described previously [29].

### 4.5. ORO Staining

Differentiated THP-1 macrophages were maintained under HG conditions for 48 h, loaded with oxLDL (100 μg/mL) for 24 h in the presence or absence of RA as indicated, washed three times with PBS and then fixed with 10% formalin for 10 min at room temperature. The fixed cells were washed three times with deionized water, stained with 0.5% ORO (Sigma-Aldrich) for 10 min at room temperature and then washed three times with deionized water. Cells were observed under a light microscope (Axiovert 40C), and images were acquired with AxioVision 4.5 software (Carl Zeiss MicroImaging GmbH, Jena, Germany).

### 4.6. Intracellular Lipid Measurement

THP-1 macrophages (1 × 10^6^) cultured under HG conditions and stimulated with oxLDL in the presence or absence of RA as described above were washed three times with ice-cold PBS and harvested in 50 µL of chloroform:isopropanol:NP-40 (200 mL, 7:11:0.1). The extract was transferred into a centrifuge tube and spun for 10 min at 15,000× *g*. The organic phase was transferred into a new tube and air-dried at 50 °C to remove the chloroform. The tube was then dried at 100 °C for 30 min to remove trace organic solvent. Total cholesterol and free cholesterol in the dried lipids were quantified using the Total Cholesterol and Cholesteryl Ester Colorimetric/Fluorometric Assay Kit (BioVision, Milpitas, CA, USA) according to the manufacturer’s instructions. Cholesteryl ester was obtained by subtracting free cholesterol from total cholesterol. The cholesterol content in the sample was calculated from the following formula: sample cholesterol concentration (C) = B/V × D (µg/µL), where B is the amount of intracellular cholesterol (µg), V is the volume of sample added to the reaction well (µL) and D is the sample dilution factor.

### 4.7. Cholesterol Efflux Assay

Macrophage-specific cholesterol efflux capacity was measured using a commercially available Cholesterol Efflux Assay Kit (Abcam, ab196985, Cambridge, MA, USA). Briefly, THP-1-derived macrophages (1 × 10^5^) were labeled with a labeling reagent which includes a fluorescent labelled cholesterol and equilibrated for 24 h in a humidified incubator at 37 °C and 5% CO_2_. Then, the media was removed, and cells were washed with fresh media and treated with RA (1, 10, 50 and 100 µM) for 24 h. Cells were added with HDL or ApoA1 cholesterol acceptors and incubated for 6 h more. Thereafter, the cholesterol contents in media and cell lysis were measured and cholesterol efflux to HDL or ApoA1 was calculated as follows:(1)$$\%\;\mathrm{cholesterol}\;\mathrm{efflux}=\frac{\mathrm{fluorescence}\;\mathrm{intensity}\;\mathrm{of}\;\mathrm{medium}}{\mathrm{fluorescence}\;\mathrm{intensity}\;\mathrm{of}\;\mathrm{cell}\;\mathrm{lysate}+\mathrm{medium}}\;\times \;100$$

### 4.8. Protein Extraction and Western Blotting

Cells were lysed using RIPA buffer (25 mM Tris-HCl, pH 7.4, 150 mM NaCl, 1% NP-40, 1% sodium deoxycholate and 0.1% SDS, 1× protease inhibitor cocktail) and centrifuged at 16,000× *g* for 20 min at 4 °C. Then, the protein concentration was quantified from the supernatant and SDS-polyacrylamide gels electrophoresis and western blot analyses were performed according to previously published procedures [26]. As primary antibodies, anti-LOX-1 (1:1000), anti-CD36 (1:1000), anti-ABCA1 (1:1000), anti-ABCG1 (1:1000), anti-phospho-STAT3 (1:500)/STAT3 (1:1000), anti-phospho-PKC (1:500)/PKC (1:1000), anti-phospho-JNK (1:500)/JNK (1:1000), anti-phospho-p38 (1:500)/p38 (1:1000), anti-phospho-ERK (1:500)/ERK (1:1000) and anti-β-actin (1:5000) antibodies were used. As a secondary antibody, a horseradish peroxidase-conjugated anti-rabbit, anti-mouse or anti-goat secondary antibody (1:5000) was used. The relative level of each protein was normalized to the level of a loading control, such as β-actin, STAT3, JNK, p38 or ERK.

### 4.9. Data and Statistical Analyses

Scanning densitometry for western blotting was performed using an Image Master® VDS system (Pharmacia Biotech Inc., San Francisco, CA, USA). One-way ANOVA was used to make statistical comparisons, followed by Tukey’s multiple comparisons test. The results were expressed as the means ± standard deviations (SDs) of at least five independent experiments, and a *p*-value < 0.05 was considered statistically significant.

## 5. Conclusions

RA might be a potential therapeutic candidate to prevent atherosclerotic cardiovascular disease complications related to diabetes.

## Figures and Tables

**Figure 1 ijms-22-08791-f001:**
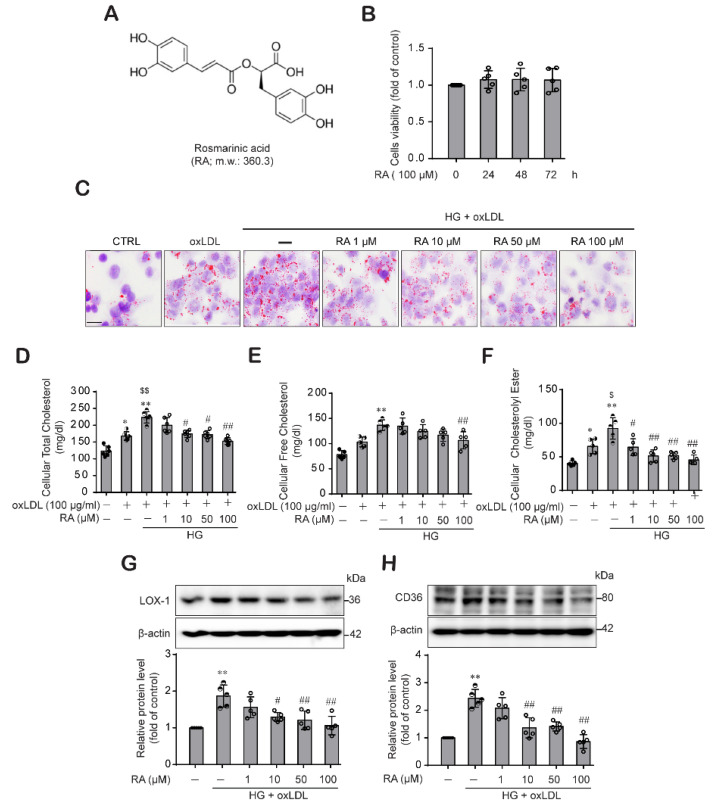
RA significantly reduced the oxLDL-induced lipid contents, cholesterol contents and levels of the SRs LOX-1 and CD36 in macrophage under HG conditions. (**A**) Chemical structure of RA. (**B**) Effect of RA on THP-1 macrophage viability. THP-1 macrophages were treated with RA (100 μM) for 24, 48 and 72 h, and then cell viability was measured by the CCK-8 assay as described in the Methods section. Results are presented as the mean ± SD from five independent experiments. (**C**) THP-1 macrophages were treated with oxLDL (100 μg/mL) under low glucose (5 mM) or HG (25 mM) conditions in the presence or absence of RA (1, 10, 50 and 100 µM). After 24 h, intracellular lipids were stained with Oil Red O (ORO) and observed under light microscopy (400× magnification) (scale bar: 50 μm). Results were confirmed by repeated experiments. (**D**–**F**) Cells were treated as described above, and intracellular total cholesterol (**D**), free cholesterol (**E**) and cholesteryl ester (**F**) were measured with a cholesterol/cholesteryl ester quantitation kit according to the manufacturer’s instructions as described in the Methods section. LOX-1 (**G**) and CD36 (**H**) protein levels were analyzed from the cell lysates by western blot analysis. The data are presented as the mean ± SD of five independent experiments. * *p* < 0.05, ** *p* < 0.01 compared to the control; ^$^ *p* < 0.05, ^$$^ *p* < 0.01 compared to the oxLDL; ^#^ *p* < 0.05, ^##^ *p* < 0.01 compared to the HG + oxLDL group.

**Figure 2 ijms-22-08791-f002:**
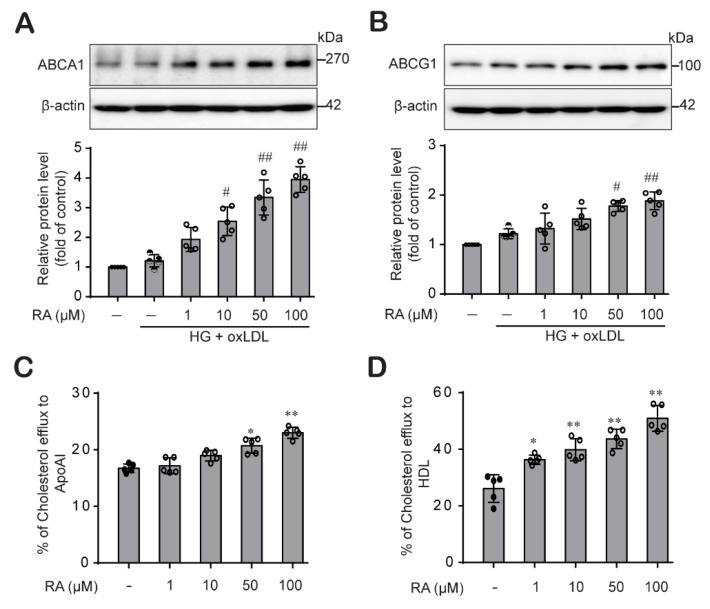
RA enhanced ABCA1 and ABCG1 protein levels and promoted cholesterol efflux in THP-1 macrophages. THP-1 macrophages cultured under HG conditions (25 mM) for 48 h were stimulated with oxLDL (100 μg/mL) for an additional 24 h in the presence or absence of RA (1, 10, 50 and 100 µM). Cell lysates were collected to determine ABCA1 (**A**) and ABCG1 (**B**) protein levels by western blot analysis. Band densities were quantified, and relative protein levels are presented as the mean ± SD of five independent experiments. (**C**,**D**) THP-1 macrophages were labeled with cholesterol and equilibrated for 24 h in the presence or absence of RA (1, 10, 50 and 100 µM). Cholesterol efflux was evaluated in THP-1 macrophages after 6 h of incubation with ApoA1 (**C**) or HDL (**D**) as described in the Methods section. The data are presented as the mean ± SD of five independent experiments. * *p* < 0.05, ** *p* < 0.01 compared to the control; ^#^ *p* < 0.05, ^##^ *p* < 0.01 compared to the HG + oxLDL group.

**Figure 3 ijms-22-08791-f003:**
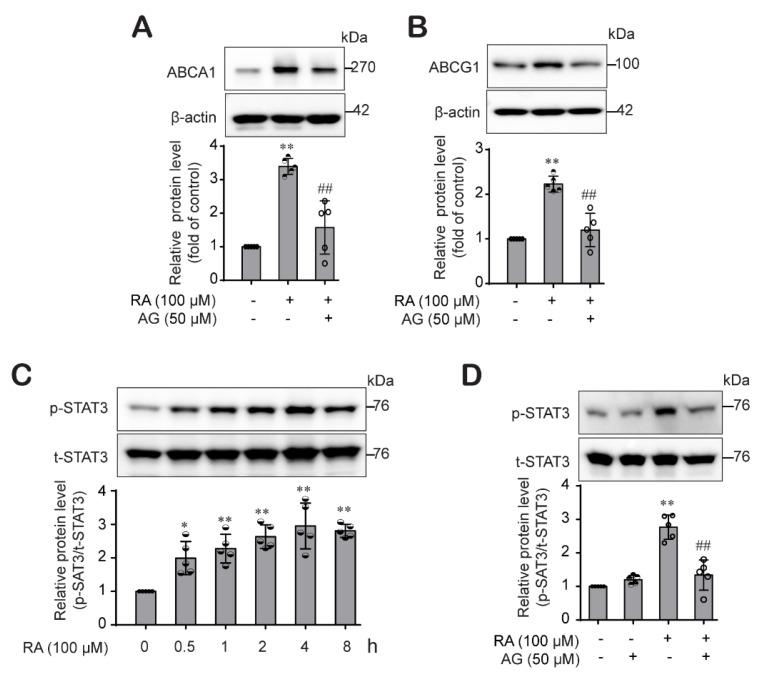
RA increased ABCA1 and ABCG1 protein expression through the JAK/STAT3 signaling pathway in THP-1 macrophages. (**A**,**B**) THP-1 macrophages were pretreated with AG490 (a JAK2 inhibitor, 50 µM) for 1 h and then treated with RA (100 µM) for 24 h. ABCA1 (**A**) and ABCG1 (**B**) protein levels were analyzed by western blot analysis. Band densities were quantified, and relative protein levels are presented as the mean ± SD of five independent experiments. (**C**) THP-1 macrophages were treated with RA (100 µM) for various time durations, after which phospho-STAT3 protein levels were determined by western blotting. (**D**) Cells were pretreated with AG490 (50 µM) for 1 h and then treated with RA (100 µM) for 4 h. The phospho-STAT3 protein levels were determined by western blotting. The data are presented as the mean ± SD of five independent experiments. * *p* < 0.05, ** *p* < 0.01 compared to the control; ^##^ *p* < 0.01 compared to the RA-treated group.

**Figure 4 ijms-22-08791-f004:**
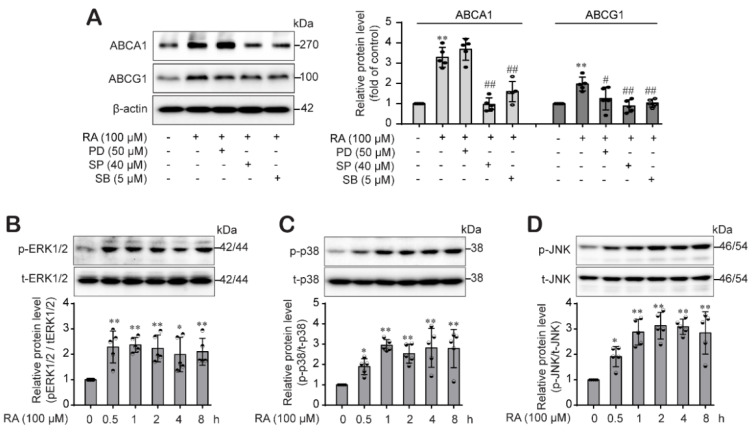
RA differentially regulates ABCA1 and ABCG1 protein expression through the MAPK signaling pathway. (**A**) THP-1 macrophages were pretreated with the MAPK inhibitors PD98059 (an ERK inhibitor, 50 μM), SB203580 (a p38 inhibitor, 40 μM) and SP600125 (a JNK inhibitor, 5 μM) (**A**) for 1 h. Then, the cells were treated with RA (100 μM) for 24 h, and ABCA1 and ABCG1 protein expression levels were analyzed by western blot analysis. (**B**–**D**) THP-1 macrophages were treated with RA (100 µM) for various durations (0.5, 1, 2, 4 and 8 h), after which phospho-ERK1/2 (**B**), phospho-p38 (**C**) and phosphor-JNK (**D**) protein levels were determined by western blotting. Band densities were quantified and normalized for the loading controls β-actin or total ERK1/2, total p38 and total JNK. Relative protein levels are presented as the mean ± SD (*n* = 5). * *p* < 0.05, ** *p* < 0.01 compared to the control; ^#^ *p* < 0.05, ^##^ *p* < 0.01 compared to the RA-treated group.

**Figure 5 ijms-22-08791-f005:**
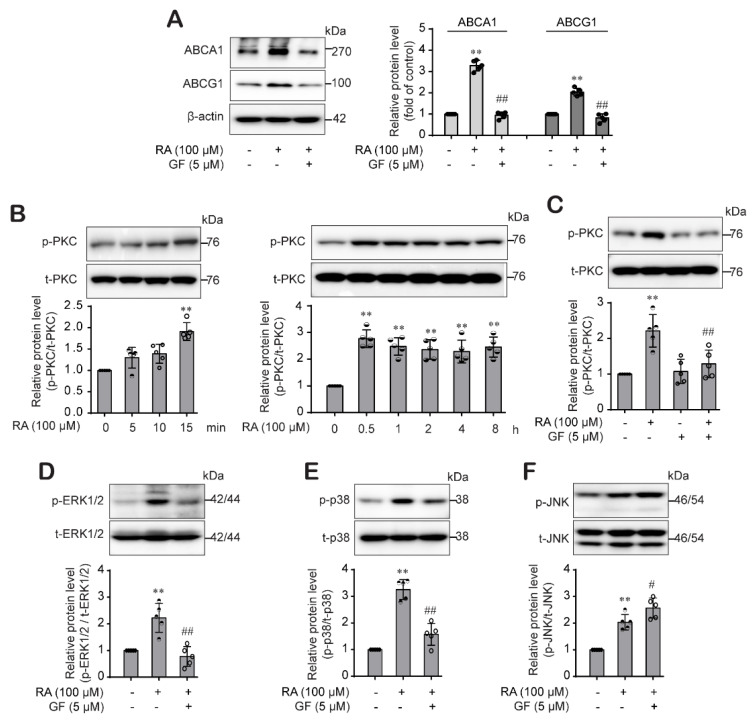
RA-activated PKC pathway was involved in the ABCA1 and ABCG1 expression by differentially regulating p38 and ERK1/2 but not the JNK pathway. (**A**) THP-1 macrophages were pretreated with GF109203X (a PKC inhibitor, 5 μM) for 30 min and then treated with RA (100) for 24 h for the determination of ABCA1 and ABCG1 protein expression by western blot analysis. (**B**) Cells were treated with RA (100 µM) in a time-dependent manner, and then phospho-PKC protein levels were determined by western blotting and normalized for the loading controls total PKC. (**C**) Cells were pretreated with GF109203X (5 µM) for 30 min and then treated with RA (100 µM) for an additional 15 min. Cell lysates were collected and phospho-PKC protein levels were determined by western blotting. (**D**–**F**) Cells were pretreated with GF109203X for 30 min and then treated with RA (100 µM) for an additional 30 min or 1 h to measure phospho-ERK1/2 protein levels or phospho-p38 and phospho-JNK protein levels, respectively, from the cell lysate by western blotting. Band densities were quantified and normalized, and relative protein levels are presented as the mean ± SD (*n* = 5). ** *p* < 0.01 compared to the control; ^#^ *p* < 0.05, ^##^ *p* < 0.01 compared to the RA-treated group.

**Figure 6 ijms-22-08791-f006:**
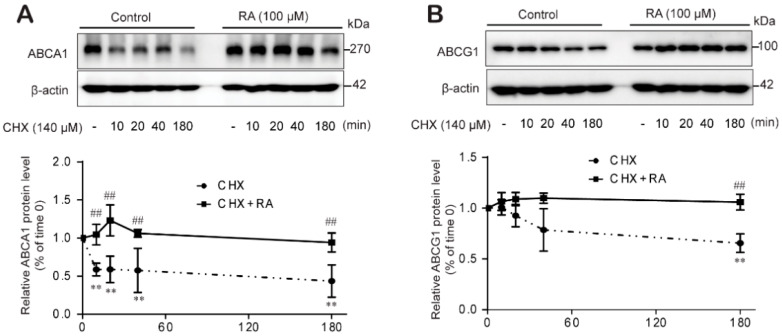
RA stabilizes ABCA1 protein levels by impairing protein degradation. THP-1 macrophages were incubated for 24 h with or without RA (100 µM) and treated with CHX (140 µM) for different durations (0, 10, 20, 40 and 180 min). Cell lysates were collected to analyze ABCA1 (**A**) and ABCG1 protein levels (**B**) by western blot analysis. Band densities were quantified, and relative protein levels are presented as the mean ± SD of five independent experiments. Significance is indicated by ** *p* < 0.01 compared with the control; ^##^ *p* < 0.01 compared to the CHK-treated group. (**C**–**E**) Cells were treated with or without RA at 100 µM for 24 h and incubated for another 3 h with the lysosomal inhibitor chloroquine (CHQ, 100 µM) (**C**), the proteasome inhibitor lactacystin (LAT, 10 µM) (**D**), or the calpain inhibitor calpeptin (CAL, 30 µg/mL) (**E**). Cell lysates were collected for ABCA1 and ABCG1 protein analysis using western blot analysis. Band densities were quantified, and relative protein levels are presented as the mean ± SD of five independent experiments. ** *p* < 0.01 compared to the control; ^##^ *p* < 0.01 compared to the CAL-treated group.

**Figure 7 ijms-22-08791-f007:**
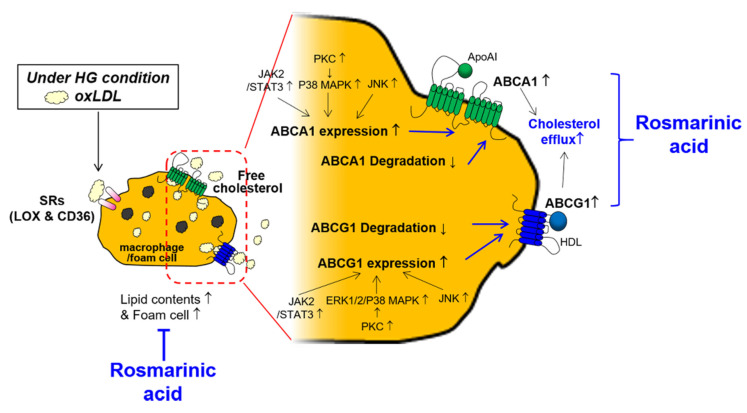
Schematic of the mechanisms by which RA reduces cellular lipid contents and increases cholesterol efflux by macrophages. Rosmarinic acid inhibits foam cells formation by decreasing lipid contents in macrophages and induces cholesterol efflux via upregulating ABC transporters. ↑; increase, ↓; decrease, T; inhibit.

## Data Availability

The data that support the findings of this study are available from the corresponding author upon reasonable request.

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
