# Peer review of "Rosmarinic Acid Increases Macrophage Cholesterol Efflux through Regulation of ABCA1 and ABCG1 in Different Mechanisms"

_ijms, 2021, doi:10.3390/ijms22168791_

Round 1

Reviewer 1 Report

In this work Nyandwi et al. studied the effect of rosmarinic acid (RA) on lipid-laden macrophages and reverse cholesterol transport in vitro. In a series of well performed in vitro experiments the authors showed that under high glucose conditions, RA reduces lipid accumulation in lipid-laden macrophages. Interestingly, Nyandwi and colleges showd that RA increase ABCA1 and ABCG1 protein levels and promotes cholesterol efflux. Mechanistically, the authors have shown that RA increase ABCA1 and ABCG1 protein levels by the activation of the JAK2/STAT3 and the MAP kinase pathway. Moreover, the authors have demonstreated that RA promotes ABCA1 and ABCG1 protein stability. The study is well presented and easy to follow, the results are interesting and fit under the scope of the journal. However, there are some limitations in the study and the authors should consider to address the next comments:

Major comments:

  • In Figure 1 the authors should show how is the lipid accumulation in low glucose conditions. To do so the authors should include a condition in where macrophages under low glucose conditions are challenged with oxLDL. ORO staining and cholesterol measurements should be provided. This result will help to understand the effect of High glucose on foam cell formation.

  • In Figure 1 the authors treat the macrophages with RA and oxLDL at the same time. This experimental design makes difficult to distinguish between the effect of RA on oxLDL uptake and cholesterol efflux. The authors should demonstrate the effect of RA on oxLDL uptake, this result will be really interesting since foam cell formation is also dependent on oxLDL uptake. Moreover, to really show the effect of RA on cholesterol efflux I would recommend the authors to treat macrophages for 24 hours with oxLDL (to generate foam cells), then remove the media and start the treatment with RA. With this experimental design the authors will avoid any uptake interference that could exist.

  • Why in Figure 2A and B is there not an increase in ABCA1 and ABCG1 protein levels after treatment with oxLDL? It is well stablished in the literature that treatment of macrophages with oxLDL increases the expression of ABCA1 and ABCG1 independently of glucose concentration in the culture media.

  • In Figure 3 the authors showed that RA increases ABCA1 and ABCG1 protein expression by JAK2/STAT3 signaling pathway. Since STAT3 is a transcription factor it would be really informative to show the mRNA levels of ABCA1 and ABCG1 under the same conditions as the western blots. The major transcription factor involve in ABCA1 and ABCG1 expression in macrophages is LXRa. Thus, to show that the RA effect on ABCA1 and ABCG1 is independent of LXRa the authors should include qRT-PCR date of other LXRa target genes like ApoE, Mertk and others.

  • The authors, base on their results (Figure 4 and 5), suggested that PKC is involve in ABCA1 and ABCG1 expression by regulation p38 and ERK1/2. However, it is not clear if RA activates PKC before p38 and ERK1/2. It would be interesting to do shorter treatments (5, 10, 15 minutes) with RA to see if PKC phosphorylation happens before p38 and ERK1/2 phosphorylation.  Also, if it is possible, a better approach than the pharmacological one, it would be to knock down PKC by siRNA and then check the effects on p38, ERK1/2 and JNK.

  • The studies on protein stability (Figure 6) should be completed with an experiment in where protein degradation is inhibited. This experiment will demonstrate if RA is interfering with protein degradation.

  • Were the experiments from figure 3 to 6 done under the same conditions (High glucose and oxLDL) than the experiments on figure 1 and 2? It is not indicated in the figure legend neither in materials and methods section. If they were done under the same conditions, please include it in the figure legend, if they were not then, why? It is critical to keep consistency specially when the authors claim that RA effect is relevant under high glucose and high lipid levels conditions.

Minor comments:

  • In the introduction it would be helpful to include a paragraph talking about the transcriptional control of ABCA1 and ABCG1 by STAT3 and LXR

  • Please rephrase line 84. The use of high glucose media does not recapitulate diabetic atherosclerosis conditions.

  • Please rephrase lines 254 to 256, the data do not support this hypothesis.

  • Please rephrase lines 271 to 273, there are not experiments in the manuscript that probe HDL functionality.

  • Please rephrase lines 285 to 287, inhibiting inhibition is difficult to understand.

  • Please rephrase lines 314 to 315, there are not experiments in the manuscript that support this affirmation.

  • It is known that activation of p38 and JNK2 in macrophages promotes ABCG1 degradation (Nagelin et al, 2009 J Biol Chem). The authors should discuss why RA increase ABCG1 protein levels despite it activates the p38 and JAK2 pathways.

  • Did the authors used L-glucose in the low glucose media to correct the difference in osmolarity with respect to the high glucose media?

Author Response

In this work Nyandwi et al. studied the effect of rosmarinic acid (RA) on lipid-laden macrophages and reverse cholesterol transport in vitro. In a series of well performed in vitro experiments the authors showed that under high glucose conditions, RA reduces lipid accumulation in lipid-laden macrophages. Interestingly, Nyandwi and colleges showed that RA increase ABCA1 and ABCG1 protein levels and promotes cholesterol efflux. Mechanistically, the authors have shown that RA increase ABCA1 and ABCG1 protein levels by the activation of the JAK2/STAT3 and the MAP kinase pathway. Moreover, the authors have demonstrated that RA promotes ABCA1 and ABCG1 protein stability. The study is well presented and easy to follow, the results are interesting and fit under the scope of the journal. However, there are some limitations in the study and the authors should consider to address the next comments:

Major comments:

  • In Figure 1 the authors should show how is the lipid accumulation in low glucose conditions. To do so the authors should include a condition in where macrophages under low glucose conditions are challenged with oxLDL. ORO staining and cholesterol measurements should be provided. This result will help to understand the effect of High glucose on foam cell formation.

-> Answer: Thank you for your suggestion.

As you recommended, we tested whether lipid is accumulated under low glucose conditions which are challenged with oxLDL. Results showed that oxLDL (100 mg/ml) treatment increased lipid accumulation and cholesterol contents (total cholesterol, free cholesterol, cholesteryl ester) in low glucose condition, which was more enhanced in high glucose condition. In the revised manuscript, we inserted this explanation and replaced Figure 1C~F with these new figures. Please see the lines 95~98 and new Figure 1C~F.

In Figure 1 the authors treat the macrophages with RA and oxLDL at the same time. This experimental design makes difficult to distinguish between the effect of RA on oxLDL uptake and cholesterol efflux. The authors should demonstrate the effect of RA on oxLDL uptake, this result will be really interesting since foam cell formation is also dependent on oxLDL uptake. Moreover, to really show the effect of RA on cholesterol efflux I would recommend the authors to treat macrophages for 24 hours with oxLDL (to generate foam cells), then remove the media and start the treatment with RA. With this experimental design the authors will avoid any uptake interference that could exist.

-> Answer: Thank you for your comments.

In Figure 1, THP-1 macrophages were treated with oxLDL (100 μg/ml) under HG (25 mM) conditions in the presence or absence of RA (1, 10, 50 and 100 µM) for 24 h. RA treatment reduced HG- and oxLDL-induced lipid contents in THP-1 macrophages.

Lipid uptake by macrophages is induced by SRs expressed on macrophages, so we evaluated the effect of RA on the key SRs responsible for oxLDL uptake by macrophages, LOX-1 and CD36. oxLDL treatment under HG conditions significantly induced LOX-1 and CD36, which were effectively reduced by RA in a dose dependent manner. From these results, we suggest that RA effectively reduced lipid contents and macrophage foam cell formation by decreasing lipid uptake which were enhanced by oxLDL under HG conditions. In the revised manuscript, we included this results as Figure 1G and H (please see new Figure 1). 

In Figure 2, we examined the effect of RA on cholesterol efflux. Because ABCA1 and ABCG1 are known as key transporters that facilitate cholesterol egress from macrophages to reduce atherosclerosis development, we first determined the effect of RA on ABCA1 and ABCG1 expression (Figure 2A, B) and then, in Figure 2C, D, we determined the effect of RA on cholesterol efflux using Cholesterol Efflux Assay Kit (Abcam, ab196985, Cambridge, MA, USA).

Because we briefly described the procedure in 4.7. Cholesterol efflux assay, there seemed to be misunderstanding. So, in the revised manuscript, we edited as follows, “Briefly, THP-1-derived macrophages (1x105) were labeled with labeling reagent which includes a fluorescent labelled cholesterol and equilibrated for 24 h in a humidified incubator at 37°C and 5% CO2. Then the media was removed, and cells were washed with fresh media and treated with RA (1, 10, 50 and 100 µM) for 24 h. Cells were added with HDL or ApoA1 cholesterol acceptors and incubated for 6 h more. Thereafter, the cholesterol contents in media and cell lysis were measured and cholesterol efflux to HDL or ApoA1 was calculated as follows” (lines 463~470).

Because RA is able to induce lipid transporters ABCA1 and ABCG1 expression even not under oxLDL and HG conditions (Figure 2A, B), RA effectively induced cholesterol efflux to HDL or ApoA1 by macrophages which were loaded with cholesterol in Figure 2C and D.  

From these results, we think that the effects of RA on oxLDL uptake (Figure 1) and cholesterol efflux (Figure 2) are distinguishable.

  • Why in Figure 2A and B is there not an increase in ABCA1 and ABCG1 protein levels after treatment with oxLDL? It is well stablished in the literature that treatment of macrophages with oxLDL increases the expression of ABCA1 and ABCG1 independently of glucose concentration in the culture media.

-> Answer: It was reported that the induction of ABCA1 and ABCG1 is repressed by high glucose stimulated macrophages (Wu et al., 2019; Mauerer et al., 2009), diabetic mice (Mauldin et al., 2006; Tang et al., 2010) and patients (Mauldin et al., 2008). Notably, Spartano et al. (2014) showed that high glucose did not change the expression of ABCA1 and ABCG1 in primary human monocytes and murine bone marrow derived macrophages.

The effect of oxLDL on the expression of ABCA1 and ABCG1 transporters poses huge controversies among researchers; some researchers have been reported that oxLDL treatment may upregulate ABCA1 and ABCG1 in macrophages (Spartano et al., 2014; Tang et al., 2004), whereas others reported opposite results showing that oxLDL downregulated ABCA1 and ABCG1 proteins in macrophages (Bekkering et al., 2014; Favari et al., 2005; Ikhlef et al., 2016).

In our results, oxLDL under HG didn’t induce the ABCA1 and ABCG1 expression, whereas RA dose-dependently increased the ABCA1 and ABCG1 expression under oxLDL and HG condition (Figure 2A,B) as well as without stimulation of oxLDL and HG (Fig. 3~6).

<References>

Wu YR, Shi XY, Ma CY, Zhang Y, Xu RX, Li JJ. Liraglutide improves lipid metabolism by enhancing cholesterol efflux associated with ABCA1 and ERK1/2 pathway. Published online 2019:1-12.

Mauerer R, Ebert S, Langmann T, Kesselschmiede Z. High glucose , unsaturated and saturated fatty acids differentially regulate expression of ATP-binding cassette transporters ABCA1 and ABCG1 in human macrophages. 2009;41(2):126-132.

Tang C, Kanter JE, Bornfeldt KE, Leboeuf RC, Oram JF. Diabetes reduces the cholesterol exporter ABCA1 in mouse macrophages and kidneys. J Lipid Res. 2010;51(7):1719-1728.

Mauldin JP, Srinivasan S, Mulya A, et al. Reduction in ABCG1 in type 2 diabetic mice increases macrophage foam cell formation. J Biol Chem. 2006;281(30):21216-21224.

Mauldin JP, Nagelin MH, Wojcik AJ, et al. Reduced expression of ATP-binding cassette transporter G1 increases cholesterol accumulation in macrophages of patients with type 2 diabetes mellitus. Circulation. 2008;117(21):2785-2792.

Spartano NL, Matthan NR, Ronxhi J, Greenberg AS, Obin MS. Lochtenstein AH. Regulation of ATP-binding cassette transporters and cholesterol efflux by glucose in primary human monocytes and murine bone marrow-derived macrophages. Exp Clin Endocrinol Diabetes

Tang CK, Yi GH, Yang JH, et al. Oxidized LDL upregulated ATP binding cassette transporter-1 in THP-1 macrophages. Acta Pharmacol Sin. 2004;25(5):581-586.

Ikhlef S, Berrougui H, Kamtchueng Simo O, Khalil A. Paraoxonase 1-treated oxLDL promotes cholesterol efflux from macrophages by stimulating the PPARγ–LXRα–ABCA1 pathway. FEBS Lett. 2016;1:1614-1629.

Bekkering S, Quintin J, Joosten LAB, Van Der Meer JWM, Netea MG, Riksen NP. Oxidized low-density lipoprotein induces long-term proinflammatory cytokine production and foam cell formation via epigenetic reprogramming of monocytes. Arterioscler Thromb Vasc Biol. 2014;34(8):1731-1738.

Favari E, Zimetti F, Bortnick AE, et al. Impaired ATP-binding cassette transporter A1-mediated sterol efflux from oxidized LDL-loaded macrophages. FEBS Lett. 2005;579(29):6537-6542.

In Figure 3 the authors showed that RA increases ABCA1 and ABCG1 protein expression by JAK2/STAT3 signaling pathway. Since STAT3 is a transcription factor it would be really informative to show the mRNA levels of ABCA1 and ABCG1 under the same conditions as the western blots. The major transcription factor involve in ABCA1 and ABCG1 expression in macrophages is LXRa. Thus, to show that the RA effect on ABCA1 and ABCG1 is independent of LXRa the authors should include qRT-PCR data of other LXRa target genes like ApoE, Mertk and others.

-> Answer: As you asked us, we performed RT-PCR (please see attached file).

Results showed that RA time dependently increased ABCA1 and ABCG1 mRNA level, and RA-induced ABCA1 and ABCG1 mRNA expression at 24 h was reduced in the presence of AG a JAK2/STAT3 inhibitor. RA also increased LXRa mRNA level and ApoE a LXRa target gene, which was decreased by AG a JAK2/STAT3 inhibitor.

As you explained here and we described in the Introduction (lines 68~74), LXRa is a most well-known transcription factor to stimulate ABCG1 and ABCA1 expression. Thus, RA also increased LXRa mRNA level and ApoE a LXRa target gene, and RA-mediated JAK2/STAT3 activation was involved in the LXRa mRNA expression and the resulting ApoE expression. Because LXRa-mediated ABCA1 and ABCG1 induction is well-known, we tried to focus on other pathway in this study, such as STAT3, PKC and MAPK pathway.

In conclusion, we suggested that RA regulates ABCA1 and ABCG1 expression levels through activation of the JAK2/STAT-3 pathway. RA-activated PKC and MAPKs are differentially involved in the effect of RA on ABCA1 and ABCG1; RA induces ABCA1 through PKC-p38 MAPK and induces ABCG1 through PKC-ERK1/2/p38 MAPK in macrophages.

  • The authors, based on their results (Figure 4 and 5), suggested that PKC is involved in ABCA1 and ABCG1 expression by regulation p38 and ERK1/2. However, it is not clear if RA activates PKC before p38 and ERK1/ It would be interesting to do shorter treatments (5, 10, 15 minutes) with RA to see if PKC phosphorylation happens before p38 and ERK1/2 phosphorylation. Also, if it is possible, a better approach than the pharmacological one, it would be to knock down PKC by siRNA and then check the effects on p38, ERK1/2 and JNK.

-> Answer: According to your suggestion, we have detected PKC phosphorylation at earlier time 5, 10, 15 min. As shown in new Figure 5B, RA activates PKC from early time at 5 min and show a significant increase from 15 min. We included this data in Figure 5B and updated the result (please see new Figure 5B and line 203).

Within the given time to us for revision, we were not able to see the effect of PKC siRNA on the p38, ERK1/2 and JNK.

  • The studies on protein stability (Figure 6) should be completed with an experiment in where protein degradation is inhibited. This experiment will demonstrate if RA is interfering with protein degradation.

-> Answer: We added the related results in new Figure 6 (please see new Figure 6 and Figure 6 legends; lines 265~271). In addition, we added the sentences as follows;

“According to reports, ABCA1 and ABCG1 protein degradation involves the proteasome-, lysosome- and calpain-mediated protein degradation pathways [30]. Therefore, we determined the protein degradation pathway through which RA regulates ABCA1 and ABCG1 protein levels using specific inhibitors of these protein degradation pathways: the proteasome inhibitor lactacystin, the lysosome inhibitor chloroquine and the calpain inhibitor calpeptin. Lactacystin, a proteasome inhibitor, and chloroquine, a lysosome inhibitor, augmented ABCA1 and ABCG1 protein levels, as did RA (Figure 6C, D). However, the protein levels of ABCA1 and ABCG1 remained unchanged upon cotreatment with either lactacystin or chloroquine and RA (Figure 6C, D), suggesting that the ability of RA to induce ABCA1 and ABCG1 protein expression is due to inhibition of the lysosome- or proteasome-mediated degradation pathways. In Figure 6E, the calpain inhibitor calpeptin increased ABCA1 and ABCG1 protein expression, similar to the effects of RA, and cotreatment with calpeptin and RA didn’t affect ABCG1 protein level, but interestingly, significantly enhanced ABCA1 protein level. These results suggest that RA stabilized ABCA1 and ABCG1 protein levels through different mechanisms; ABCA1 by impairing proteasome- and lysosome-mediated degradation and ABCG1 by impairing proteasome-, lysosome- and calpain-mediated degradation in THP-1 macrophages.” (lines 241~258)

  • Were the experiments from figure 3 to 6 done under the same conditions (High glucose and oxLDL) than the experiments on figure 1 and 2? It is not indicated in the figure legend neither in materials and methods section. If they were done under the same conditions, please include it in the figure legend, if they were not then, why? It is critical to keep consistency specially when the authors claim that RA effect is relevant under high glucose and high lipid levels conditions.

-> Answer: As we explain before, RA induces ABCA1 and ABCG1 expression regardless of oxLDL and HG. From Figure 3 to Figure 6, we examined the mechanisms by which RA induces ABCA1 and ABCG1. Therefore, we treated macrophages with RA not under oxLDL and HG condition.

Minor comments:

  • In the introduction it would be helpful to include a paragraph talking about the transcriptional control of ABCA1 and ABCG1 by STAT3 and LXR

-> Answer: We added a paragraph describing the transcriptional regulation of ABCA1 and ABCG1 by STAT3 and LXR as follows,

“Regarding to the ABCA1 and ABCG1 induction mechanisms, liver X receptor (LXR) agonists are most well-identified transcription factors to stimulate ABCG1 and ABCA1 expression, resulting in promotion of cholesterol efflux from macrophages, and eventually protection against atherosclerosis in mice [15,23]. Moreover, it has been known that Janus kinase (JAK) activated signal transducer and activator of transcription 3 (STAT3) signaling pathway increased ABCA1 and ABCG1 levels in macrophages [24,25].” Please see lines 68~74.

  • Please rephrase line 84. The use of high glucose media does not recapitulate diabetic atherosclerosis conditions.

-> Answer: According to your suggestion, we rephrased line 84 as follows,RA could reduce lipid contents and cholesterol level in macrophages treated with oxLDL under high glucose conditions” (lines 92, 93)

  • Please rephrase lines 254 to 256, the data do not support this hypothesis.

-> Answer: Thank you for your suggestion. During revision, we rephrased that paragraph. Please see lines 280~302.

  • Please rephrase lines 271 to 273, there are not experiments in the manuscript that probe HDL functionality.

-> Answer: According to your recommendation, we deleted our assumption and rephrased the paragraph based on our findings. Please see lines 303~315.

  • Please rephrase lines 285 to 287, inhibiting inhibition is difficult to understand.

-> Answer: We corrected that sentence as follows; “Moreover, it has been reported that HSP70 downregulates ABCA1 and ABCG1 expression through inhibiting JNK and its downstream Elk-1 in macrophages [46].” Please see lines 327-329.

  • Please rephrase lines 314 to 315, there are not experiments in the manuscript that support this affirmation.

-> Answer: We rephrased as follows, “RA reduces atherosclerotic foam cells and enhances cholesterol efflux from macrophages under HG conditions by inducing the transporters ABCA1 and ABCG1.” Please see lines 375~377.

  • It is known that activation of p38 and JNK2 in macrophages promotes ABCG1 degradation (Nagelin et al, 2009 J Biol Chem). The authors should discuss why RA increase ABCG1 protein levels despite it activates the p38 and JAK2 pathways.

-> Answer: Thank you for your kind comments.

In Nagelin et al., 2009, they reported that 12/15-lipoxygnease targets ABCG1 for serine phosphorylation and destabilization through p38- and JNK2-dependent pathways in murine macrophage. We carefully hypothesize that the difference between our results and the previous work may be partly due to the use of different species of cells. Our results have been done in human macrophages. Indeed, murine and human cells possess different ABCG1 isoforms, which present distinct post-translational processing. It is reported that ABCG1 (+12) isoform is expressed in human cells but not in mouse cells (Gelissen et al., 2010). Notably, ABCG1 (+12) phosphorylation stabilized ABCG1 protein levels in human cells and facilitated ABCG1-dependent cholesterol efflux to HDL (Gelissen et al., 2012; Watanabe et al., 2019).

One more possible reason is that different stimuli may cause different outcomes at different time points, even if they come into contact with the same signaling molecule. In our previous study, oxLDL under HG condition activated p38 phosphorylation, which were involved in the inflammasome activation (Nyandwi et al., 2020), but oxLDL stimulation under HG condition didn’t induce ABCA1 and ABCG1 expression in this study. However, RA activated p38 phosphorylation and significantly induced ABCA1 and ABCG1 expression through p38 pathway in this study. Thus, further investigation to elucidate the role of p38 and JNK pathways in human ABCG1 protein degradation is needed.

We included this discussion in the lines 342~360.

<References>

Nagelin MH, Srinivasan S, Nadler JL, Hedrick CC. Murine 12/15-lipoxygenase regulates ATP-binding cassette transporter G1 protein degradation through p38- and JNK2-dependent pathways. J Biol Chem. 2009, 284, 31303-31314.

Gelissen IC, Cartland S, Brown AJ, et al. Expression and stability of two isoforms of ABCG1 in human vascular cells. Atherosclerosis 2010, 208, 75-82.

Watanabe T, Kioka N, Ueda K, Matsuo M. Phosphorylation by protein kinase C stabilizes ABCG1 and increases cholesterol efflux. J. Biochem. 2019, 166, 309-315.

Gelissen IC, Sharpe LJ, Sandoval C, et al. Protein kinase A modulates the activity of a major human isoform of ABCG1. J Lipid Res. 2012;53(10):2133-2140.

Nyandwi JB, Ko YS, Jin H, Yun SP, Park SW, Kim HJ. Rosmarinic acid inhibits oxLDL-induced inflammasome activation under high-glucose conditions through downregulating the p38-FOXO1-TXNIP pathway. Biochem. Pharmacol. 2020, 182,114246.

  • Did the authors used L-glucose in the low glucose media to correct the difference in osmolarity with respect to the high glucose media?

-> Answer: As control condition (a low glucose condition), we used low glucose DMEM media containing 1 g/L of D-Glucose (5 mM Glu) and 20 mM mannitol to correct the osmolarity. As HG condition, we used high glucose DMEM media containing 4.5 g/L of D-Glucose (25 mM Glu). To help the readers’ understanding we added this point. Please see lines 428, 429.

Reviewer 2 Report

1. Some grammars and writings should be modified. Here are some examples in the abstract and introduction EX. in line 24, HG ??? the first time of abbreviation should be noted, hyperglycemia? EX. in line 24, macrophages EX line 43-44, where they take up cholesterol ….. EX, line 47, CD36 or lectin-like… EX. line 51, and following metabolism by the liver. EX. atherosclerotic cardiovascular development. 2. In Figure 3, phosphorylation of STAT under the inhibition AG490 (a JAK2 inhibitor) should be presented at least by Western blot study. 3. In Figure 5, which PKC was examined in the Western blot study? Conventional? Novel? Atypical? phosphorylation of PKC under the inhibition GF109203X should be presented at least by Western blot study. 4. At least the authors need to discuss possible downstream mechanisms after MAPK activation on ABC expression. Are ABC proteins target genes by MAPK activated transcriptional factors?

Author Response

1. Some grammars and writings should be modified. Here are some examples in the abstract and introduction

EX, in line 24, HG ??? the first time of abbreviation should be noted, hyperglycemia?

EX, in line 24, macrophages

EX, line 43-44, where they take up cholesterol …..

EX, line 47, CD36 or lectin-like…

EX, line 51, and following metabolism by the liver.

EX, atherosclerotic cardiovascular development.

-> Answer: Thank you for your kind comments. We reviewed the manuscript and tried to correct typos and writings. Specifically, we corrected what you indicated. Please see the parts you pointed out.

2. In Figure 3, phosphorylation of STAT under the inhibition AG490 (a JAK2 inhibitor) should be presented at least by Western blot study.

-> Answer: We did as you suggested and presented in the Figure 3D. In the revised manuscript, we replaced Figure 3 with new Figure 3 and updated the result and the Figure 3 legends. Please see new Figure 3, lines 160, 161; 171~173.

3. In Figure 5, which PKC was examined in the Western blot study? Conventional? Novel? Atypical? phosphorylation of PKC under the inhibition GF109203X should be presented at least by Western blot study.

-> Answer: We used commercial antibody against PKC from Santa Cruz biotechnology (cat # sc-10800) as we described in the 4.1. Materials (lines 406~408).

In addition, as you suggested, we presented phosphorylation of PKC under the inhibition GF109203X in the Figure 5C and updated the Figure legends (lines 223-226).

4. At least the authors need to discuss possible downstream mechanisms after MAPK activation on ABC expression. Are ABC proteins target genes by MAPK activated transcriptional factors?

-> Answer: According to your suggestion, we included the following sentences, “Several important studies demonstrated that MAPK-mediated activation of downstream transcription factors resulted in the induction of ABCA1 and ABCG1 in macrophages. Tanshinone IIA activated ERK/Nrf2/HO-1 signaling pathway to mediate ABCA1 and ABCG1 induction in macrophages [47]. Similarly, p38 MAPK-activated sp1 interacted with LXRα to activate ABCA1 transcription in macrophages [45]. Moreover, the activation of the JNK signaling pathway stimulated activator protein-1 (AP-1) DNA binding activity to upregulate ABCA1 expression in macrophages [48].” Please see lines 338~344.

Reviewer 3 Report

The aim of the study was to investigate the effect of rosmarinic acid on the efflux of cholesterol from macrophages. The authors showed that rosmarinic acid regulates the work of transporters (ABCA1, ABCG1) responsible for removing lipids from macrophages. In addition, the authors explained the possibility of activation of various signaling pathways by RA.
The article was written concisely and clearly. The results of the study provide new information on the possibility of regression of atherosclerotic changes by substances of natural origin.

Nevertheless, I have a few comments:
1. The concentration of oxLDL (100 µM) used in the work is high. Typically a concentration of 50 µM is used. Has oxLDL toxicity to macrophages been tested?
2. # 309 - The authors did not investigate the effect of rosmarinic acid on atherosclerosis in vivo. In this case, we can only talk about the potential role of rosmarinic acid in reducing atherosclerotic foam cells.
3. It should also be added at what stage of atherosclerosis regression of atherosclerotic changes is possible.
4. There are no data on the bioavailability of rosmarinic acid. In what form can it be used?
5. Please explain "HG" at the beginning of the article.

Author Response

The aim of the study was to investigate the effect of rosmarinic acid on the efflux of cholesterol from macrophages. The authors showed that rosmarinic acid regulates the work of transporters (ABCA1, ABCG1) responsible for removing lipids from macrophages. In addition, the authors explained the possibility of activation of various signaling pathways by RA. The article was written concisely and clearly. The results of the study provide new information on the possibility of regression of atherosclerotic changes by substances of natural origin.

Nevertheless, I have a few comments:

1. The concentration of oxLDL (100 µM) used in the work is high. Typically a concentration of 50 µM is used. Has oxLDL toxicity to macrophages been tested?

-> Answer: In our experimental system using endothelial cells, we used to use 100 mg/ml oxLDL and it has no cytotoxicity. In this study using macrophage, it seemed to have no cytotoxicity. During revision, we prove out that 100 mg/ml oxLDL has also no cytotoxicity in macrophages. Please see the attached file.

2. # 309 - The authors did not investigate the effect of rosmarinic acid on atherosclerosis in vivo. In this case, we can only talk about the potential role of rosmarinic acid in reducing atherosclerotic foam cells.

-> Answer: Thank you your suggestion. As you recommended, we changed the sentence as follows, “Taken together, these findings suggest for the first time that RA reduces atherosclerotic foam cells and enhances cholesterol efflux from macrophages under HG conditions by inducing the transporters ABCA1 and ABCG1, and RA differentially regulate ABCA and ABCG1 (Figure 7).” Please see lines 377~380.

3. It should also be added at what stage of atherosclerosis regression of atherosclerotic changes is possible.

-> Answer: Thank you for your suggestion. According to your suggestion, we added sentences as follows, “Atherogenesis is a complex process that progresses from the development of fatty streak to the formation of thrombus within the intima. Lipids uptake by macrophages and foam cells formation and subsequent fatty streak formation occur at early stage of atherogenesis (Moore & Tabas, 2011). Moreover, defective cholesterol efflux from macrophages lead to free cholesterol-induced cytotoxicity and may promote macrophage death in advanced lesions (Barrett, 2020). Given that RA modulates macrophage foam cells formation and increased cholesterol efflux, RA would be beneficial at early stages of atherogenesis. Further clinical studies are needed.” Please see lines 383~391.

<References>

Moore KJ, Tabas I. Macrophages in the pathogenesis of atherosclerosis. Cell. 2011;145(3):341-355. doi:10.1016/j.cell.2011.04.005

Barrett TJ. Macrophages in Atherosclerosis Regression. Arterioscler Thromb Vasc Biol. 2020;40(1):20-33. doi:10.1161/ATVBAHA.119.312802

4. There are no data on the bioavailability of rosmarinic acid. In what form can it be used?

-> Answer: Some researchers have studied the pharmacokinetic profile of RA in animal model and human study. According to Baba et al. (2004), when the rat is administered with RA 50 mg/kg, the concentration of RA in plasma reached Cmax 4.64 mmol/L at 30 min and its metabolites methyl-RA and m-coumaric acid were detected 5.03 mmol/L at 1 h and 0.75 mmol/L at 8 h, respectively. In human study (Baba et al., 2005), 6 male healthy volunteers who took 200 mg RA showed Cmax 1.15 ± 0.28 mmol/L at 30 min in plasma. When they were fasted and took 250 mg and 500 mg RA, the Cmax was significantly increased into 72.22 ± 12.01 mmol/L at 1 h and 162.20 ± 40.20 mmol/L at 1 h, respectively (Noguchi-Shinohara et al., 2015).

From these reports, we can guess that the bioavailability of RA by oral is not good but 100 mM of RA that exerts effectiveness in our study could be achievable in human. Moreover, in vivo, the metabolites of RA can work as well. In addition, we have performed preliminary experiments to first confirm whether RA affect lipids profile in wild type C57BL/6 mice. We fed C57BL/6 mice with a high fat diet (60% kcal fat) for 8 weeks and treated RA 50 mg/kg and 100 mg/kg in suspended form in water for additional 8 weeks. Strikingly, we observed that RA treatment (50 mg/kg and 100 mg/kg; it can reach plasma concentration probably 5 mmol/L ~ 10 mmol/L from the references) significantly decreased mice plasma triglycerides and total plasma cholesterol (please see attached file). From these results, we can suggest that the doses of RA (50 mg/kg and 100 mg/kg) are effective to reduce plasma cholesterol.

<References>

Baba S, Osakabe N, Natsume M, Terao J.Life Sci. 2004 May 28;75(2):165-78. Orally administered rosmarinic acid is present as the conjugated and/or methylated forms in plasma, and is degraded and metabolized to conjugated forms of caffeic acid, ferulic acid and m-coumaric acid.

Baba S, Osakabe N, Natsume M, Yasuda A, Muto Y, Hiyoshi K, Takano H, Yoshikawa T, Terao J. Absorption, metabolism, degradation and urinary excretion of rosmarinic acid after intake of Perilla frutescens extract in humans. Eur J Nutr 2005 Feb;44(1):1-9.

Noguchi-Shinohara M, Ono K, Hamaguchi T., Iwasa K, Nagai T, Kobayashi S, Nakamura H, Yamada M. Pharmacokinetics, Safety and Tolerability of Melissa officinalis Extract which Contained Rosmarinic Acid in Healthy Individuals: A Randomized Controlled Trial. PLoS One. 2015 May 15;10(5):e0126422.

5. Please explain "HG" at the beginning of the article.

-> Answer: We explained HG as high glucose. Please see lines 24 and 87.

Round 2

Reviewer 1 Report

I would like to thank the authors for the effort that they made to improve their work. The authors have addressed all the concerns in a satisfactory way.

Reviewer 2 Report

The responses are accepted